# Filamin protects myofibrils from contractile damage through changes in its mechanosensory region

Lucas A. B. Fisher[1], Belén Carriquí-Madroñal[2], Tiara Mulder[3], Sven Huelsmann[2], Frieder Schöck[1]*, Nicanor González-Morales[3]*

1 Department of Biology, McGill University, Montreal, Quebec, Canada, 2 Interfaculty Institute of Cell Biology, Universität Tübingen, Tübingen, Germany, 3 Department of Biology, Dalhousie University, Halifax, Nova Scotia, Canada

* nicanor.gonzalez@dal.ca (NG-M); frieder.schoeck@mcgill.ca (FS)

**Data Availability Statement:** Drosophila strains and plasmids are available upon request. All data necessary for confirming the conclusions of the

## Abstract

Filamins are mechanosensitive actin crosslinking proteins that organize the actin cytoskeleton in a variety of shapes and tissues. In muscles, filamin crosslinks actin filaments from opposing sarcomeres, the smallest contractile units of muscles. This happens at the Z-disc, the actin-organizing center of sarcomeres. In flies and vertebrates, filamin mutations lead to fragile muscles that appear ruptured, suggesting filamin helps counteract muscle rupturing during muscle contractions by providing elastic support and/or through signaling. An elastic region at the C-terminus of filamin is called the mechanosensitive region and has been proposed to sense and counteract contractile damage. Here we use molecularly defined mutants and microscopy analysis of the Drosophila indirect flight muscles to investigate the molecular details by which filamin provides cohesion to the Z-disc. We made novel filamin mutations affecting the C-terminal region to interrogate the mechanosensitive region and detected three Z-disc phenotypes: dissociation of actin filaments, Z-disc rupture, and Z-disc enlargement. We tested a constitutively closed filamin mutant, which prevents the elastic changes in the mechanosensitive region and results in ruptured Z-discs, and a constitutively open mutant which has the opposite elastic effect on the mechanosensitive region and gives rise to enlarged Z-discs. Finally, we show that muscle contraction is required for Z-disc rupture. We propose that filamin senses myofibril damage by elastic changes in its mechanosensory region, stabilizes the Z-disc, and counteracts contractile damage at the Z-disc.

## Author summary

Muscles work by contracting and relaxing in response to signals from the nervous system. Muscles are made up of long, slender cells called muscle fibers. Inside these fibers are even smaller units called sarcomeres, which are responsible for the actual contraction of the muscle. The sliding of actin and myosin filaments shortens the sarcomeres, causing the muscles to contract. As sarcomeres contract, the entire muscle shortens, pulling on the

article are present within the article, figures, and tables.

**Funding:** This work was supported by operating grants MOP-142475 to FS and PJT-155995 to FS from the Canadian Institutes of Health Research and by RGPIN-02984-2022 to NGM from the Natural Sciences and Engineering Research Council of Canada. The funders had no role in study design, data collection and analysis, decision to publish, or preparation of the manuscript.

**Competing interests:** The authors have declared that no competing interests exist.

tendons attached to it. This pulling action is what creates movement. Sarcomeres however are prone to contractile damage and so they have proteins to hold the sarcomere together. One of these proteins is filamin, a large protein which has a special elastic part called the mechanosensitive region, which might be crucial for providing stability to the sarcomeres as they contract. To test this hypothesis, we made versions of filamin with increased or decreased elasticity and studied how muscles behaved with these changes. We find that when filamin lost its elasticity, sarcomeres are more prone to breaking while when filamin has extra elasticity, the sarcomeres do not break and instead form large protein aggregates which we think represent the sarcomere response to contractile damage. This research gives us a better understanding of how our muscles sustain contractile damage, which is important for everyone from athletes to people recovering from injuries.

## Introduction

Striated muscles are big cells that in addition to the typical cellular constituents are characterized by large cytoskeletal cables called myofibrils which are composed of repeating subunits called sarcomeres [1]. Sarcomeres are responsible for the contraction of muscles and are made up of actin and myosin filaments. The Z-disc is a thin structure that defines the boundaries of the sarcomere and serves as the anchor point for actin filaments [2]. The M-line is at the center of the sarcomere where myosin filaments are anchored [2]. Myosin heads slide actin filaments towards the center shortening the sarcomere to provide the basis for muscle contraction [3]. The Z-disc is also where the myofibril diameter is set [4], and it is a hub for metabolic enzymes [5].

Concentric contractions occur when the muscle fibers shorten in response to a load or resistance. Eccentric contractions occur when the muscle fibers lengthen while still under tension. Eccentric contractions generate more force than concentric contractions but cause myofibril damage [6]. The eccentric myofibril damage is characterized by a series of structural steps that start with the overstretching and breaking of the weakest Z-disc in the myofibril [7]. Then extensive Z-disc remodeling follows including the widening of the Z-disc [8]. Finally, the Z-disc structure is reassembled. During this process, some Z-disc proteins, including Filamin, Xin, Hsp70, and αB-crystallin [9–11], accumulate at the lesions and are thought to mediate Z-disc stability and repair [9–11]. Filamin is one of the earliest proteins to localize at damaged regions and a commonly used marker of myofibrillar damage [10,12].

Filamin is a large protein consisting of an elongated structure with a length of approximately 80 nm [13,14]. Human filamins are composed of two identical subunits, each of which contains an N-terminal actin-binding region composed of two Calponin Homology (CH) domains, followed by 24 immunoglobulin-like (Ig) domains, the last Ig domain being a dimerization domain [13]. The Ig domains are arranged into a flexible rod-like structure that allows filamin to interact with a wide variety of binding partners, including membrane receptors, signaling proteins, and other cytoskeletal proteins. Humans have three filamin genes *Filamin-a*, *Filamin-b*, and *Filamin-c*. Structurally, Filamin-a, -b, and -c are highly similar, but their expression patterns vary. Filamin-a is the most abundant and widely distributed member [13]. Filamin-b shows ubiquitous expression with elevated levels in endothelial cells and chondrocytes [15]. Filamin-c is primarily expressed in adult muscles [13,16]. Drosophila has a single filamin gene called *cheerio* that replaces the three vertebrate filamins [17]. Here we refer to cheerio as filamin. The structure of Drosophila filamin is identical to the vertebrate filamins with the exception that is has only 22 Ig domains instead of 24. In muscles, filamin localizes to

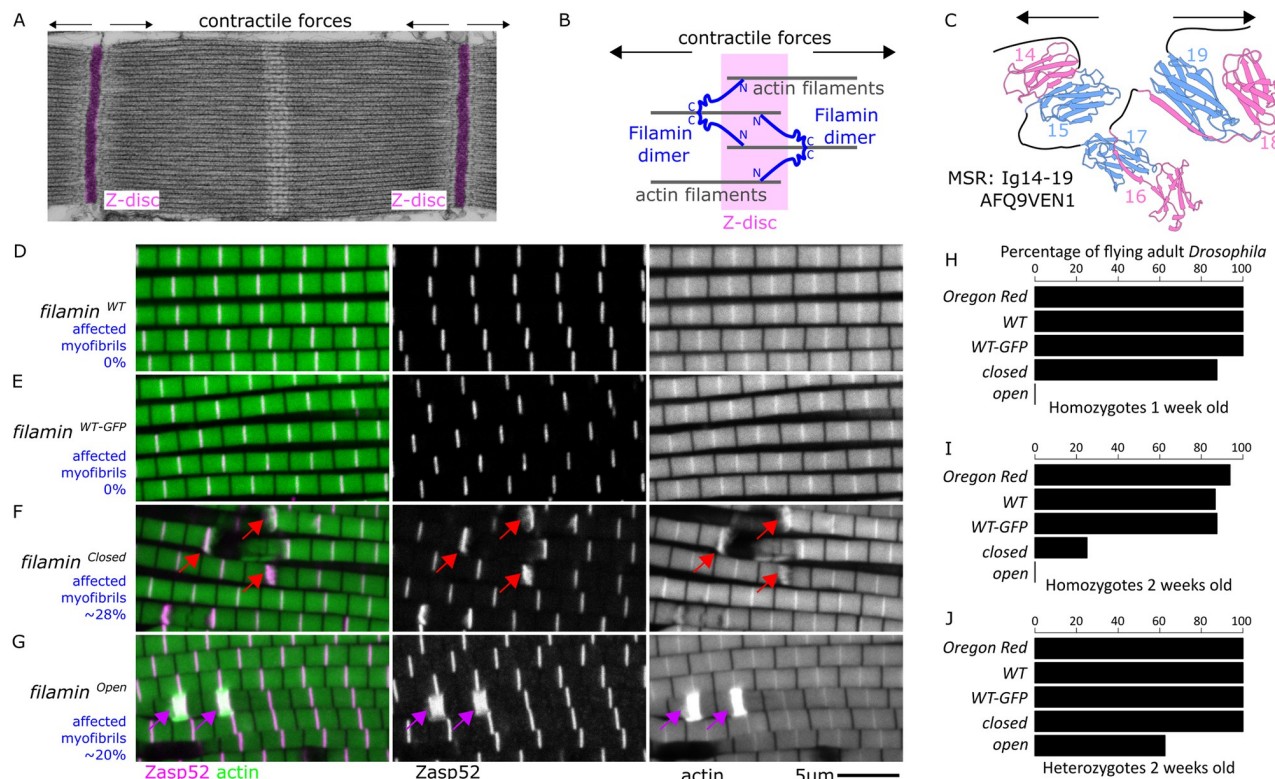

**Fig 1. Filamin constitutive open and closed mutants have distinct myofibril defects.** A) A transmission electron microscope image of a sarcomere; the Z-discs are highlighted in magenta, and arrows point towards the direction in which actin filaments are pulled during the contraction cycle. B) Cartoon model of filamin at the Z-disc. The dimerization domain at the C-terminus points away from the Z-disc. C) AlphaFold model of the mechanosensitive region of Drosophila filamin (AFQ9VEN1). D-G) Confocal microscopy images of different filamin mutant conditions. Filamin alleles were analyzed over the *Df(3R)Ex6176* deletion mutant. Actin filaments are shown in green and Zasp52-mCherry in magenta. Individual channels are shown in grayscale. D) *filamin^WT* does not have any obvious muscle defect. E) *filamin^WT-GFP* also does not have any obvious muscle defect, suggesting that the C-terminal GFP tag does not affect filamin function. F) In *filamin^closed-GFP* mutants, the myofibril appears ruptured at the Z-disc, marked with Zasp52-mCherry (red arrows). A slight increase in Zasp52 levels occurs at the breaking points. G) In *filamin^open-GFP* mutants the myofibrils are not ruptured but have occasional enlarged Z-discs (purple arrows). The scale bar is 5 μm. H-J) Flight percentage plots for homozygotes at 1 week (H) and 2 weeks (I), as well as for heterozygotes at 2 weeks (J). In panels H-J, N = 30 flies for each data bar.

the Z-disc and to the myotendinous junctions in both vertebrates and Drosophila [18–21]. Filamin is organized into the Z-disc with its actin-binding domains at the center of the Z-disc, where they bind actin filaments from opposing sarcomeres, and the dimerization domain at the periphery of the Z-disc (Fig 1A and 1B; [22,23]). This organization is parallel to the contractile forces of muscles. In cells, actin filaments are crosslinked at variable angles from 35 to 90 degrees [24]. In contrast, at the Z-discs actin filaments are parallel to each other and filamin connects the actin filaments from two opposing sarcomeres [25].

Filamin is sensitive to mechanical pulling forces [26], which makes it ideal for sensing Z-disc stretching damage and initiating repair signals. The mechanosensitive region (MSR) of Drosophila filamin consists of the Ig domains 14–19, which adopts a globular structure (Fig 1C; [27]). The MSR is organized into three Ig domain pairs, in Drosophila these are Ig 14–15, 16–17, and 18–19. Interaction sites between the contact faces of Ig 16 and 17, and another one between Ig 18 and 19 keep the MSR in a closed inactive state [28]. However, each of these interactions is sensitive to mechanical pulling forces and upon pulling forces of 3–5 pN, the intradomain interactions are lost and the Ig domains are free to interact with other proteins

[14,26,28]. The MSR is in a closed inactive state when its domain pairs are tightly bound or in an open active state when its domain pairs are free to bind interacting proteins.

Here we use novel filamin mutants to analyze the effect of the MSR on myofibril ultrastructure. We focused on the indirect flight muscles (IFM) because of their regular sarcomere pattern [25,29,30]. We found that locking the MSR of filamin into a closed state leads to Z-disc rupturing while locking the MSR into an open state leads to enlarged Z-discs, which we interpret as a compensatory mechanism for the sarcomere damage that occurs during contractions to stabilize the Z-disc and prevent it from breaking. We found that the removal of the entire MSR region leads to a ruptured disc and myofibril disintegration. We found that the Ig domain pairs 16–17 and 18–19 are required for Z-disc stability. We found that filamin dimerization is also required for Z-disc cohesion. Finally, we found that muscle contraction is required for myofibril rupture in filamin mutants. Our data provides a detailed analysis of the domains of filamin involved in myofibril stability and damage sensing, and we provide new tools to study filamin in Drosophila.

## Methods

### Drosophila stocks and maintenance

Drosophila stocks were cultured and maintained on standard cornmeal glucose media. Zasp52-mCherry is a gene trap made by swapping the $Zasp52^{MI02988}$ MIMIC transposon with an artificial exon containing the mCherry coding sequence [4,31]. $Cher^{s24-attp}$ contains an attp landing site, is on a *white* mutant background, and is used as the founder line for all the cher knock-in mutants [27]. The replacement alleles $cher^{WT}$, $cher^{WT-GFP}$, $cher^{open-GFP}$, $cher^{closed-GFP}$ $cher^{ΔIg14-21-GFP}$ were described previously [27]. The *Df(3R)Ex6176* deficiency uncovers the filamin gene and was generated through FLP-induced recombination between two FRT-carrying transgenic insertions at 3R:17,053,662 and 17,149,107 [32]. S1 Data contains a list of the full genotypes that were used in the main figures.

### Generation of cher/filamin alleles

The deletion mutants are derived from *pGE-attBGMR-cher$^{WT-GFP}$*, which contains the last five exons of cher cloned from BAC RP98-2L16 with a C-terminal mGFP6 tag cloned into pGE-attB-GMR (DGRC Stock 1295; RRID: DGRC_1295) via EcoRI and KpnI sites [27]. DNA fragments lacking specific domains were generated by overlap extension PCR and subcloned by conventional cloning into pGE-attB-GMR-cher240WT-GFP replacing wild type sequences. The resulting plasmids were sequence-verified and integrated into the attP site of $cher^{s24-attp}$. The correct integration was confirmed by PCR; the w+ marker was removed using Cre recombinase [33]. The aligned amino acid sequences of all mutant forms used in this study are available as a supplementary document; proteins were aligned using the ClustalW function from MacVector (version 18.0.2). The transcription start site for the cher90 isoform has been omitted from the rescue construct, resulting in the absence of the small cher90 isoform. Since GFP is positioned at the C-terminus, all other isoforms are tagged.

### Western blot analysis of filamin mutants

Western blotting experiments from adult flies were done as previously [30]. Briefly, 20 adult flies were homogenized in 100 μl of 2x SDS running buffer via mortar and pestle, boiled at 100˚C for 10 minutes, spun down, and then the supernatant with all the soluble proteins was collected. The soluble fraction was run on a NuPage (Thermo Fisher) Tris-Acetate gel and transferred onto a nitrocellulose membrane. Rabbit anti-GFP antibody (Chromotek) was then

added at a 1:5000 dilution concentration, followed by anti-Rabbit ECL (Millipore) at 1:10000 concentration. Membranes were scanned with a C-DiGit Blot Scanner (Licor). As loading control, we used Ponceau staining (5% glacial acetic acid and 0.1% Ponceau S), followed by washing with 1x Tris-Buffered Saline 0.1% Tween.

## Muscle staining and microscopy

Dissection and microscopy experiments were done following our previous protocol [5,31,34–36]. Briefly, thoraces from female flies were cut in half longitudinally using a sharp blade, then fixed for 1 h at room temperature using 4% formaldehyde. Following three washes with PBS triton 0.1%, the muscles were further dissected to remove them from the thoracic cuticle. The isolated muscles were then incubated in PBS with 1:1000 488-Phalloidin or 555-Phalloidin (Acti-stain, Cytoskeleton Inc.) to stain the actin filaments. After washing with PBS triton 0.1% again, the muscles were mounted in Mowiol 4–88 mounting media (Sigma 9002-89-5). To image the muscle ultrastructure, we used a Leica TCS SP8 Confocal Microscope. Images were taken with an HC PL APO 63x oil NA = 1.4 objective. We used a 488 nm, 20 mW, AOTF laser for GFP and 488-Phalloidin and a 552 nm, 20 mW laser for mCherry and 555-phalloidin. Only the outer layer of myofibrils was used because phalloidin does not penetrate well into the tissue. Imaging settings were kept identical within experiments. Well-stained muscles were selected and aligned using the 63x objective without any additional digital zoom. A large area at the center of the muscles was selected. Then, randomly selected areas within this area were imaged at 9x digital zoom with a pixel resolution of 1024x1024. At least 10 muscles were analyzed per condition. Finally, we cropped an area of the scanned image to show the representative phenotypes. To calculate the ratios of affected myofibrils, we used the uncropped images that typically contain 10–15 myofibrils. The ratio of affected myofibrils is the number of affected myofibrils in an image divided by the total number of myofibrils in the same image. S1 Data contains a summary table of myofibril phenotypes and S3 Data contains the myofibril counts. To measure GFP intensity, we utilized the following ImageJ script, which creates a 250-pixel circle around a selected spot and calculates the average gray value.

```
macro "Macro enlarge measure [a]" {run("Enlarge...", "enlarge = 4 pixel"); run("Measure");}
```

## Immobilization experiments

Newly eclosed flies were placed in a custom-made immobilization chamber with a diameter of 6.35 mm and a length of 5 cm. In this chamber, flies were able to walk along the chamber but had limited room, which prevented flight. After 13–15 days in the chamber at 25°C their IFM were dissected and imaged. Simultaneously, flies of the same genotype were kept in normal vials as the flying group.

## Behavioral assays

One-week-old flies were anesthetized on an ice pad. Then a 10 μl pipette tip was glued to their thoraces using fast-drying transparent nail polish. The flies were placed in the middle of the sensor using a 3D printable micromanipulator [37]. Flies were left to recover for 5 minutes at room temperature. After careful placement of the flies in the middle of the two detectors, we recorded the reflected light from both wings for periods of 30 seconds. Flies were stimulated to start flight by directing a gentle puff of air at them. Plots and correlation coefficients were calculated using custom R scripts. The double-wing sensor is composed of two QRD1114 sensors (DigiKey QRD1114) connected to an Arduino UNO microcontroller. The two QRD1114

sensors are mounted into a custom-made 3D-printed holder. The maximum reflectance signal occurs when the wing is 0.635 mm from the sensor. 3 mm is the maximum range at which the sensors detect something. The wing movement was recorded in tethered flies. The data was analyzed in R software using custom scripts. Briefly, the output of the Arduino was saved as a CSV file containing two columns, one per wing. We selected a 5-millisecond time window randomly, plotted the results, and calculated the Pearson correlation value of the wing movements. We repeated the process 50 times and counted the number of synchronous and asynchronous events. We analyzed 10 flies per genotype and recorded 1 minute of continuous flying. We used a threshold value of r > 0.5 to classify synchronized and unsynchronized flying events. S3 Data contains supporting information for myofibril phenotype data, wing beat synchronization data, and GFP intensity data. For flight assays, 30 flies were released individually from a plastic vial: if they flew upwards, they were marked as a flyer, whereas if they fell or glided to the ground they were marked as a non-flyer.

## Results

### Mechanosensitivity of filamins is required for preventing myofibril rupture

When filamin is removed from the IFM using a filamin-RNAi transgene, it results in the presence of ruptured and frayed myofibrils (S1 Fig and [18]). To test if the mechanical signaling property of filamin is involved in this phenotype, we used two well-characterized mutants that affect specifically the mechanical opening of the filamin mechanosensory region (MSR). The wildtype filamin MSR switches between open and closed states. The *filamin*$^{open-GFP}$ mutation prevents the interactions between Ig 16 and Ig 17 and between Ig 18 and Ig 19 which shifts the balance toward an open state and presumably keeps the MSR always open. The *filamin*$^{closed-GFP}$ mutant has the opposite effect, it makes the interactions between these domain pairs stronger and therefore shifts the balance towards a closed conformation [27]. As controls we used *filamin*$^{WT-GFP}$ and *filamin*$^{WT}$ which were made in the same way as the other mutants but using the wildtype sequence of filamin with or without a C-terminal GFP sequence.

To standardize the genetic background and introduce a Z-disc fluorescence marker, we first generated a line carrying the *Df(3R)Ex6176* deficiency, which completely removes the filamin gene, and a *Zasp52-mCherry* allele that robustly labels the Z-disc. We crossed this line with the filamin mutants and analyzed the IFM of the progeny carrying a filamin mutant over *Df(3R)Ex6176* and a copy of Zasp52-mCherry. Filamin myopathy patients show late-onset muscle function loss [16,38], indicating filamin's role in muscle maintenance. In Drosophila, filamin mutants fly initially but lose flight ability around day 15 [39]. Because of the suspected role of filamin in muscle maintenance, we used 15-day-old flies. Muscles from the control line *filamin*$^{WT}$ did not show any obvious phenotype (Fig 1D), suggesting that the filamin replacement technique does not affect muscle structure. The other control line *filamin*$^{WT-GFP}$ likewise did not show any muscle defects (Fig 1E), suggesting that the addition of a GFP tag does not have any detrimental effect on the function of filamin in muscles. In contrast, muscles from the *filamin*$^{closed-GFP}$ mutant had several ruptured myofibrils (Fig 1F). The ruptures occurred always at Z-discs, marked by Zasp52-mCherry, and only in a few Z-discs per myofibril (Fig 1F). These ruptures are reminiscent of the filamin-RNAi phenotype described previously (S1 Fig, [18]). We did not find ruptured myofibrils in the *filamin*$^{open-GFP}$ mutant muscles but instead, we found many actin aggregates that occur always at the Z-discs, marked by Zasp52-mCherry (Fig 1G). Because these aggregates occur always at the Z-disc and because they suggest an over-recruitment of Z-disc proteins, we refer to them as enlarged Z-discs. The *filamin*$^{open-GFP}$ mutants are entirely flightless during weeks 1 and 2 after eclosion (Fig 1H and 1I), and their muscles do not have any ruptured Z-discs. In

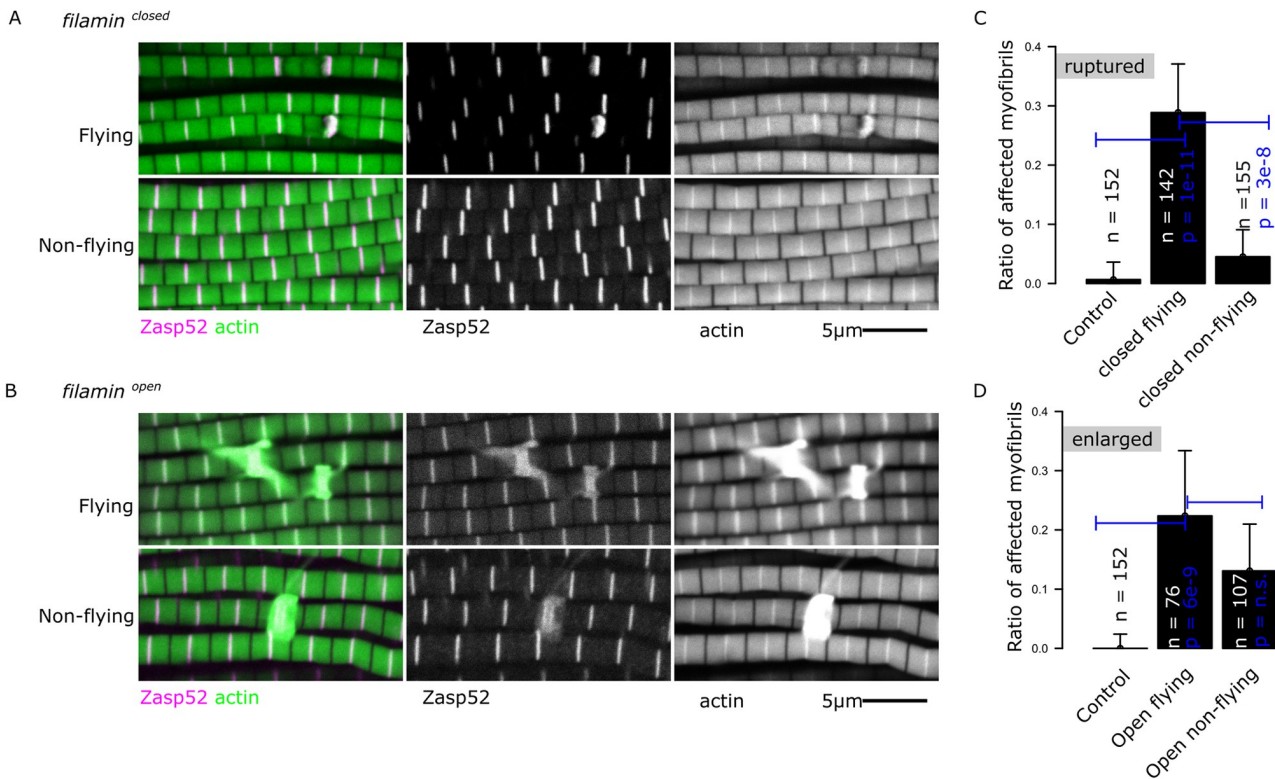

**Fig 2. Muscle contractions underlie the rupture defects in *filamin*^*closed-GFP*^ mutants.** A) Confocal microscopy images of *filamin*^*closed-GFP*^ mutants with normal flying (upper panel) or with restricted flying (lower panel); *filamin*^*closed-GFP*^ mutants with restricted flying do not develop ruptured myofibrils. B) Confocal microscopy images of *filamin*^*open-GFP*^ mutants with normal flying (upper panel) or with restricted flying (lower panel); both conditions develop enlarged Z-discs. Filamin alleles were analyzed over the *Df(3R)Ex6176* deletion mutant. Actin filaments are shown in green and Zasp52-mCherry in magenta. C) Plot of the number of ruptured myofibrils in *filamin*^*closed-GFP*^ mutants in flying and non-flying conditions. D) Plot of the number of myofibrils with enlarged Z-discs in *filamin*^*open-GFP*^ mutants in flying and non-flying conditions over the total number of myofibrils. In C and D, confidence intervals were calculated at 95 by an exact binomial test, and the p-values using a 2-sample test for equality of proportions with continuity correction. N.S., not significant (p-value = 0.15 in D).

contrast, the *filamin*^*closed-GFP*^ mutants can fly during the first week but then become progressively flightless during the second week (Fig 1H and 1I). Heterozygote *filamin*^*closed-GFP*^ mutants fly as well as controls (Fig 1J), suggesting that a single working copy of filamin is enough for complete rescue.

Since the ruptured myofibril phenotype in the *filamin*^*closed-GFP*^ mutants resembles the myofibril damage occurring during muscle contraction in humans [8,40], we hypothesized that muscle contractions are required for the myofibrils to break in these filamin mutants. To test this hypothesis, we analyzed the effect of blocking muscle contractions on the structure of muscles in filamin mutants. We kept flies for 15 days in very thin tubes with a 6.35 mm diameter, which prevent the flies from flying, and then analyzed their muscles using confocal microscopy. The *filamin*^*closed-GFP*^ mutants that were prevented from flying had normal-looking muscles, whereas the ones that were allowed to fly had ruptured Z-discs (Fig 2A and 2C). In contrast, the ratio of enlarged Z-discs in the *filamin*^*open-GFP*^ mutants was only minimally reduced (Fig 2B and 2D). Our results suggest that muscle contractions are required for the Z-disc rupture phenotype but not for the enlarged Z-disc phenotype.

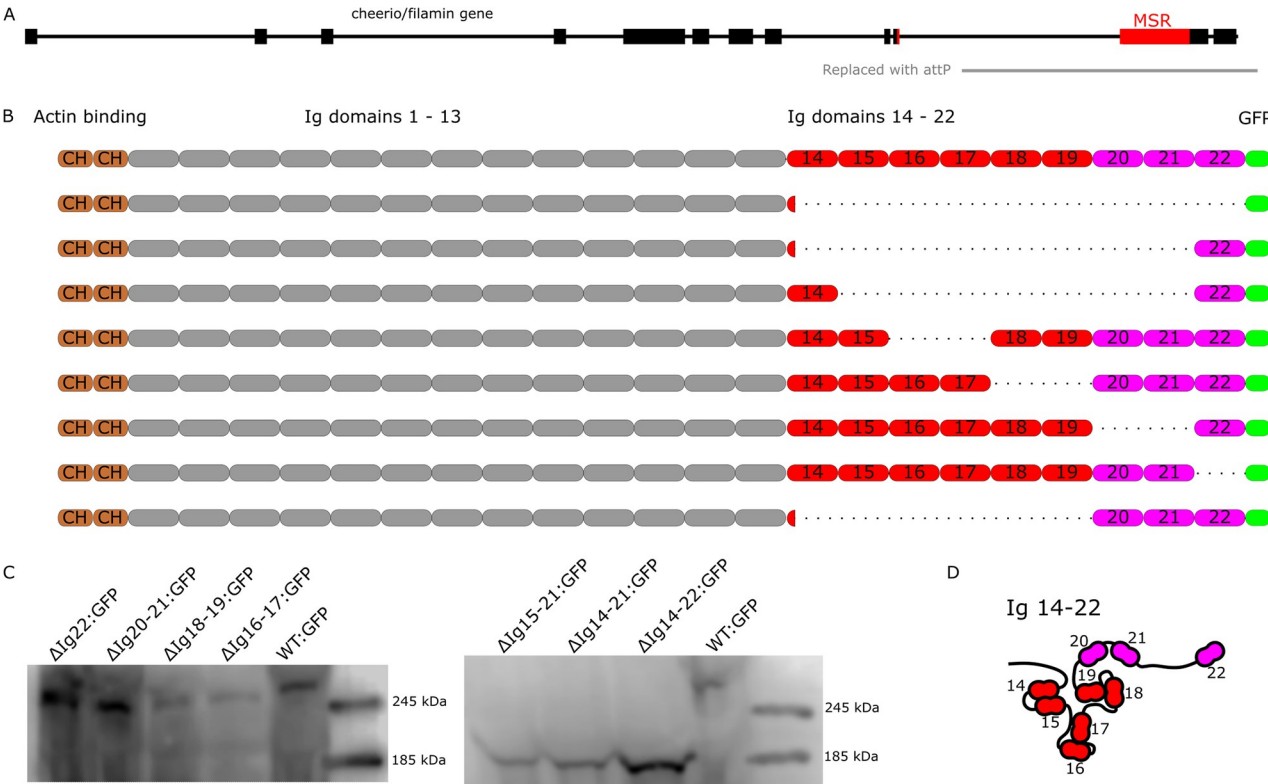

**Fig 3. Generation of novel filamin alleles tagged with a C-terminal GFP.** A) Cartoon of the cheerio/filamin locus. Boxes represent exons. The red boxes denote the exons that encode the MSR. The *filamin^s24* mutant is a deletion of the C-terminal part of the filamin gene replaced with an attB Integrase landing site (gray line). B) Illustration of the deletion mutants generated. The MSR is colored in red and Ig domains 20–22 in purple The rows in descending order illustrate *filamin^WT-GFP*, *filamin^ΔIg14-22-GFP*, *filamin^ΔIg14-21-GFP*, *filamin^ΔIg15-21-GFP*, *filamin^ΔIg16-17-GFP*, *filamin^ΔIg18-19-GFP*, *filamin^ΔIg20-21-GFP*, *filamin^ΔIg22-GFP*, and *filamin^ΔIg14-19-GFP*, which was generated previously [27]. C) Western blots showing the molecular weights of the truncated filamin proteins using an anti-GFP antibody from thorax lysates. All the deletions have the correct predicted molecular weights, *filamin^ΔIg15-21-GFP* 175 kDa, *filamin^ΔIg14-21-GFP* 185 kDa, *filamin^ΔIg14-22-GFP* 197 kDa, *filamin^ΔIg22-GFP* 257 kDa, *filamin^ΔIg20-21-GFP* 247 kDa, *filamin^ΔIg18-19-GFP* 248 kDa, *filamin^ΔIg16-17-GFP* 247 kDa, *filamin^WT-GFP* 267 kDa. D) Cartoon of the Ig domains 14–22. The MSR is colored in red and the following Ig domains 20–22 in purple. The interactions between the Ig domains 14 with 15, 16 with 17, and 18 with 19 are the structural basis of the MSR function.

## Genomic engineering a set of small deletion mutants tagged with GFP

To reveal how Drosophila filamin mediates its function during muscle maintenance, we performed a structure-function analysis of the mechanosensitive C-terminus. We generated 7 new lines (Fig 3A and 3B, and S2 Data), each with a small deletion that removes specific Ig domains of the C-terminus, using a knock in approach [27]. We also included in our analysis *filamin^ΔIg14-19-GFP* that lacks the entire MSR [27]. Western blot analysis of these filamin forms showed that each line expressed a filamin protein of the expected size (Figs 3C and S2). The resulting constructs were verified by sequencing and are incorporated into the *filamin* gene using *filamin^s24-attp* as the landing site. We made 8 mutants: *filamin^ΔIg14-22-GFP*, *filamin^ΔIg14-21-GFP*, *filamin^ΔIg15-21-GFP*, *filamin^ΔIg16-17-GFP*, *filamin^ΔIg18-19-GFP*, *filamin^ΔIg20-21-GFP*, and *filamin^ΔIg22-GFP*. The transcription start site for the cher90 isoform has been omitted from the rescue construct, resulting in the absence of the small cher90 isoform [27]. Since GFP is positioned at the C-terminus, all other isoforms are tagged.

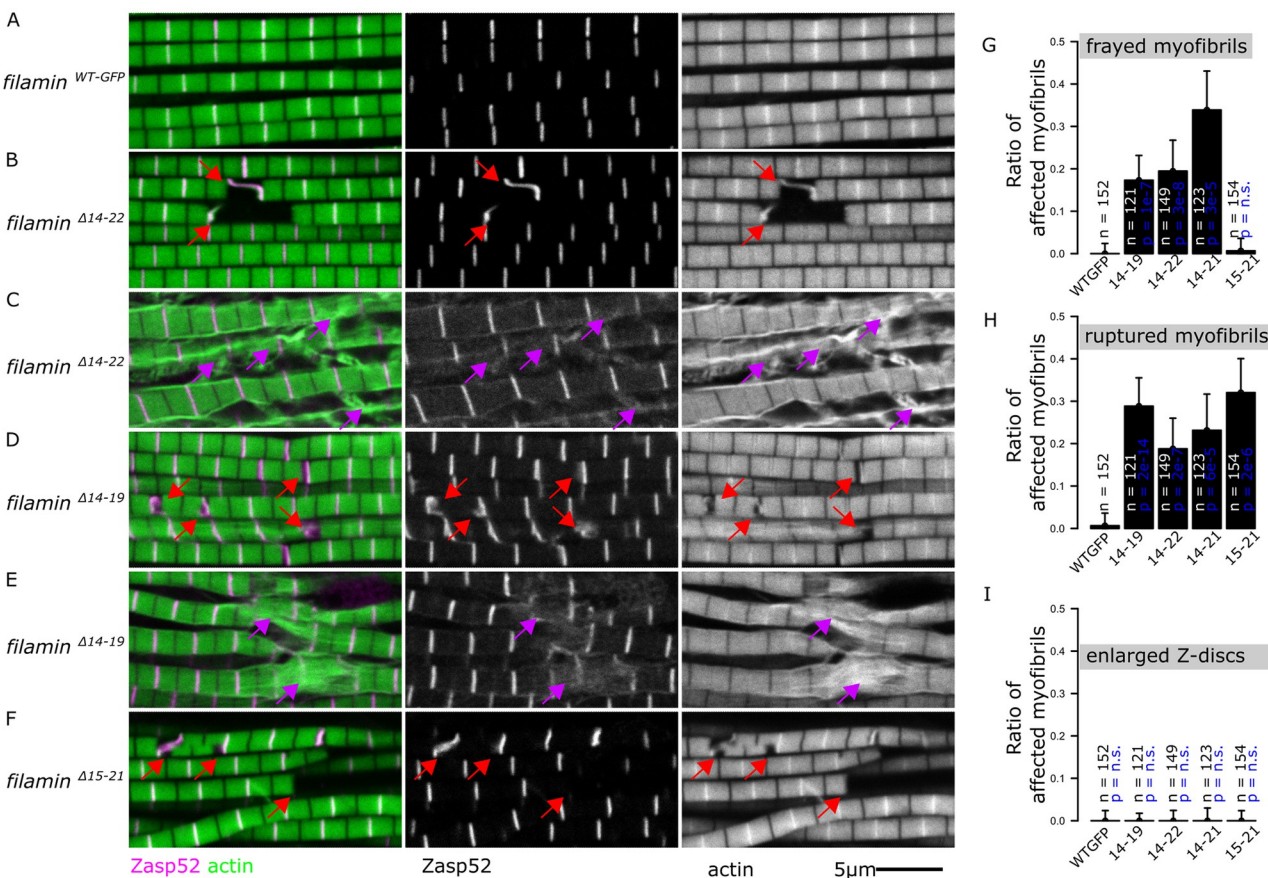

**Fig 4. Large deletion mutants have ruptured and frayed myofibrils.** A-F) Confocal microscopy images of *filamin* mutant muscles showing ruptured or frayed myofibrils. Actin filaments are shown in green and Zasp52-mCherry in magenta. Filamin mutants were analyzed over the *Df(3R)Ex6176* deletion mutant. Red arrows point at ruptured myofibrils. Purple arrows point to frayed myofibrils. The scale bar is 5 μm. A) Normal myofibrils in *filamin$^{WT-GFP}$*. B) Ruptured myofibrils in *filamin$^{ΔIg14-22-GFP}$* mutant. C) Frayed myofibrils in *filamin$^{ΔIg14-22-GFP}$* mutant. D) Ruptured myofibrils in *filamin$^{ΔIg14-19-GFP}$* mutant. E) Frayed myofibrils in *filamin$^{ΔIg14-19-GFP}$* mutant. F) Ruptured myofibrils in *filamin$^{ΔIg15-21-GFP}$* mutant. G) Plot of the ratio of frayed myofibrils in different large deletions mutants. Frayed myofibrils are not present in the *filamin$^{ΔIg15-21-GFP}$* mutant. H) Plot of the ratio of ruptured myofibrils over the total number of myofibrils in different large deletions mutants. I) Plot of the ratio of myofibrils with enlarged Z-discs in different large deletions mutants. Confidence intervals were calculated at 95 by an exact binomial test, and the p-values using a 2-sample test for equality of proportions with continuity correction.

## Large deletion mutants have ruptured Z-discs and myofibril fraying

We first analyzed the IFM phenotypes of the large deletion mutants over the *Df(3R)Ex6176* deficiency chromosome by confocal microscopy. We started with *filamin$^{ΔIg14-22-GFP}$*, which removes the entire C-terminal region starting at Ig 14. In contrast to the regular myofibrils present in *filamin$^{WT-GFP}$* and *filamin$^{WT}$* (Fig 4A), the *filamin$^{ΔIg14-22-GFP}$* deletion mutant exhibited ruptured myofibrils and sometimes actin filaments detaching or fraying away from the myofibrils (Fig 4B and 4C, quantified in 4G and 4H). In the other large deletion mutants, *filamin$^{ΔIg14-19-GFP}$* (Fig 4D and 4E, quantified in 4G and 4H) and *filamin$^{ΔIg14-21-GFP}$* (Figs 4G and 4H, S3), we also observed ruptured and frayed myofibrils. Fraying and ruptured myofibrils were observed in the same muscle or even the same myofibril (S3 Fig). Finally, we tested *filamin$^{s24-attp}$*, the original landing mutant, which removes the entire region starting from Ig 14 and lacks a stop codon. Rupturing Z-discs and fraying myofibrils are common in the *filamin$^{s24-attp}$* mutant muscles (S3 Fig). We then tested if preventing muscle contractions would rescue

the ruptured and the frayed phenotypes in the *filamin*$^{\Delta Ig14-22-GFP}$ deletion mutant. We observed a reduction in the occurrence of ruptured myofibrils in flies that were restricted from flying. However, the proportion of frayed myofibrils remained consistent between flying and non-flying flies (S3 Fig). In contrast to the defects from the big deletion mutants that affect Ig 14 (Fig 4C–4E), *filamin*$^{\Delta Ig15-21-GFP}$ had ruptured but not frayed myofibrils (Fig 4F, quantified in G and H), suggesting that Ig 14 may be crucial for preventing actin detachment. Neither the open nor the closed mutants resulted in frayed myofibrils, suggesting that the mechanical switch of the MSR is not required to prevent actin filament fraying. Finally, none of the MSR deletion mutants had enlarged Z-discs (Fig 4I).

## Filamin dimerization is required to prevent Z-disc rupturing

To crosslink actin filaments, actin-binding proteins such as filamin must have at least two actin-binding domains. A single filamin monomer has only one actin-binding domain and is not able to crosslink actin filaments. The *filamin*$^{\Delta Ig22-GFP}$ allele removes specifically the dimerization domain and allows us to interrogate the requirement of forming filamin dimers. The sarcomeres of the *filamin*$^{\Delta Ig22-GFP}$ mutant muscles exhibited clear ruptured Z-discs, although no fraying of the myofibrils was observed (Fig 5A and 5B). We then investigated whether preventing muscle contractions would ameliorate the *filamin*$^{\Delta Ig22-GFP}$ mutant phenotype. Similarly to *filamin*$^{closed-GFP}$ flies, the ruptured phenotype of the *filamin*$^{\Delta Ig22-GFP}$ was rescued by inhibiting flight (S3C Fig). Overall, these findings suggest that dimerization is necessary to protect the Z-disc from contractile damage, but not required to prevent actin filaments fraying from the myofibril.

## Within the MSR, both mechanosensitive dimers contribute to mechanosensing and filamin lacking Ig 20–21 behaves like *filamin* $^{open-GFP}$

We then analyzed the muscle phenotypes of three small deletions that remove single-domain pairs, filamin Ig domains 16–17 and Ig 18–19, which are part of the MSR, and Ig domains 20–21, which are just outside the MSR (Fig 3B and 3D). Muscles from the *filamin*$^{\Delta Ig16-17-GFP}$ and the *filamin*$^{\Delta Ig18-19-GFP}$ mutants had ruptured myofibrils (Fig 5C and 5D). Overall, these mutants behave as *filamin* $^{closed-GFP}$ mutants (Figs 1 and 2).

Unlike all the other deletion mutants, the muscles from the *filamin*$^{\Delta Ig20-21-GFP}$ mutant flies have enlarged Z-discs, like the ones observed in *filamin* $^{open-GFP}$ (Fig 5E). We measured the size of the enlarged Z-discs using Zasp52-mCherry in the two mutants and noticed that they have identical sizes (S4 Fig). Actin accumulates at very high levels in the enlarged Z-discs and covers a slightly wider area compared to Zasp52 (5E Fig). However, we noted that unlike in *filamin* $^{open-GFP}$ mutants, the *filamin*$^{\Delta Ig20-21-GFP}$ homozygote mutants can fly, suggesting that the flightless defect observed in *filamin* $^{open-GFP}$ mutants is not linked to the enlarged Z-discs. Finally, we counted the number of ruptured and enlarged myofibrils in all the small deletion mutants and noted that the mutants would either have ruptured myofibrils or enlarged Z-discs but not both (Fig 5F and 5G), which is identical to what we observed in the open and closed mutants. In contrast to the large mutants, the single-domain pair mutants show limited fraying (Fig 5H).

## Mutant filamin forms localize to the Z-disc

We used the GFP tag in all the filamin mutants to test the domains required for Z-disc localization. Because the fluorescence from the endogenous filamin is low in the IFM Z-discs, we used the homozygote flies, which have both filamin copies tagged with GFP. Given that the myofibril phenotypes in the mutants are not present in all the sarcomeres, we analyzed normal-

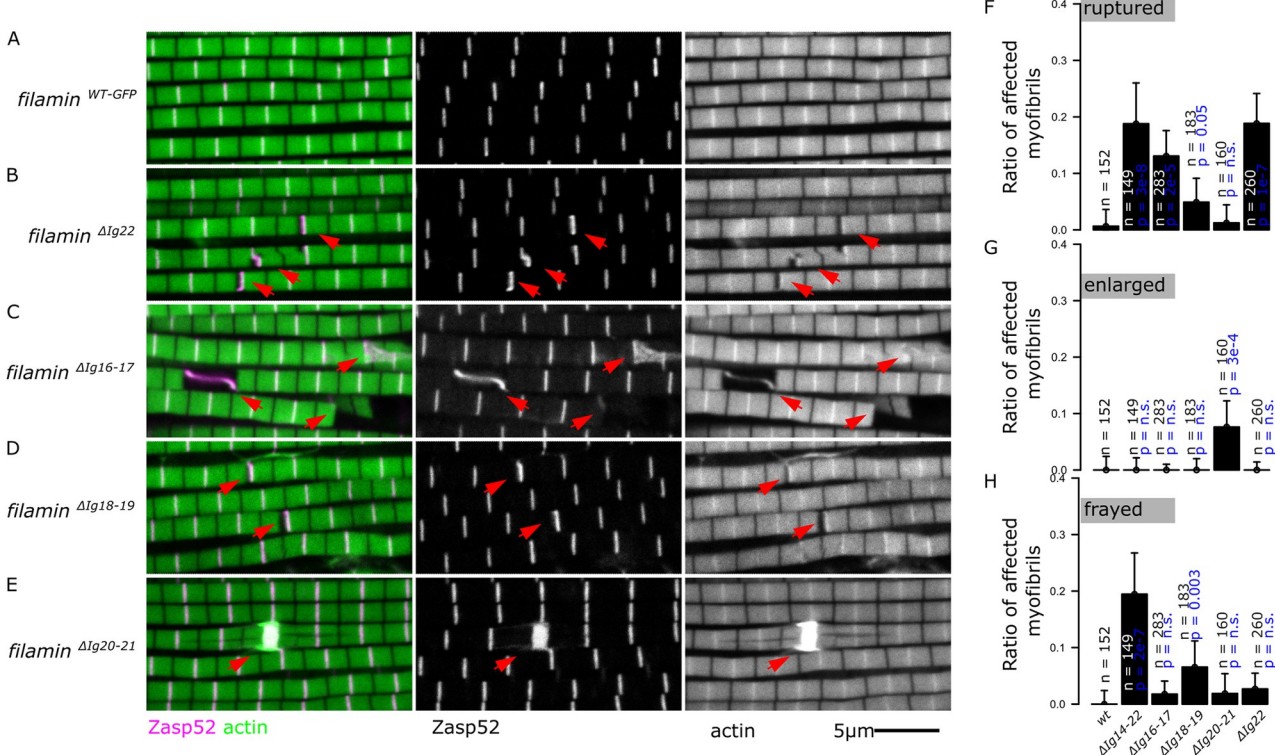

**Fig 5. Mutants lacking one pair of Ig domains, or the dimerization domain have distinct myofibril phenotypes.** A-E) Confocal microscopy images of diverse *filamin* mutant muscles. Filamin alleles were analyzed over the *Df(3R)Ex6176* deletion mutant. Actin filaments are shown in green and Zasp52-mCherry in magenta. A) Example of normal myofibrils. B) Example of ruptured myofibrils present in the dimerization domain mutant *filamin^{Δ22-GFP}* (red arrows). C) Example of ruptured myofibrils present in the *filamin^{ΔIg16-17-GFP}* mutant (red arrows). D) Example of ruptured myofibrils present in the *filamin^{ΔIg18-19-GFP}* mutant (red arrows). E) Myofibrils with enlarged Z-discs present in the *filamin^{Δ20-21-GFP}* mutant. In this mutant, the myofibrils do not break. F) Plot of the ratio of ruptured myofibrils over the total number of myofibrils. All single-domain pair mutants have ruptured myofibrils except for the *filamin^{Δ20-21-GFP}* mutant. G) Plot of the ratio of enlarged Z-discs over the total number of myofibrils. Only the *filamin^{Δ20-21-GFP}* mutant has enlarged Z-discs. H) Plot of the ratio of frayed myofibrils. Confidence intervals were calculated at 95 by an exact binomial test, and the p-values using a 2-sample test for equality of proportions with continuity correction.

looking sarcomeres, ruptured sarcomeres, and enlarged Z-discs. In the control wild-type, filamin localizes to the Z-disc and diffusely to the cytoplasm (Fig 6A, red asterisks). The *filamin-^{closed-GFP}* mutant also localizes to the Z-disc in normal-looking myofibrils (Fig 6B, red asterisks). In ruptured myofibrils, however, it accumulates at the ruptured Z-disc (Fig 6C and 6H). The *filamin^{open-GFP}* form mainly localizes to the enlarged Z-discs and the GFP signal from normal-looking Z-disc is weak but often detectable (Fig 6D and 6I). Interestingly, the signal from the Z-discs adjacent to the enlarged discs diminishes, suggesting that filamin molecules relocate from normal Z-discs into the enlarged ones (Fig 6J).

The largest deletion mutant we have, *filamin^{Δ14-22-GFP}*, has a clear Z-disc localization pattern in normal-looking myofibrils (Fig 6F), suggesting that the region responsible for Z-disc localization is in the N-terminal region anywhere between the CH-domains and Ig 13. The filamin mutant form lacking the entire MSR, *filamin^{Δ14-19-GFP}*, accumulates at the ruptured Z-discs (Fig 6E). We then measured the intensity of GFP signal at individual Z-discs. Even though all the mutants localized to the Z-disc, all of them except for *filamin^{Δ16-17-GFP}* and *filamin^{Δ18-19-GFP}* had a diminished Z-disc signal compared to the control (Fig 6G). The accumulation of filamin in ruptured discs was similar among mutants that together span the entire C-

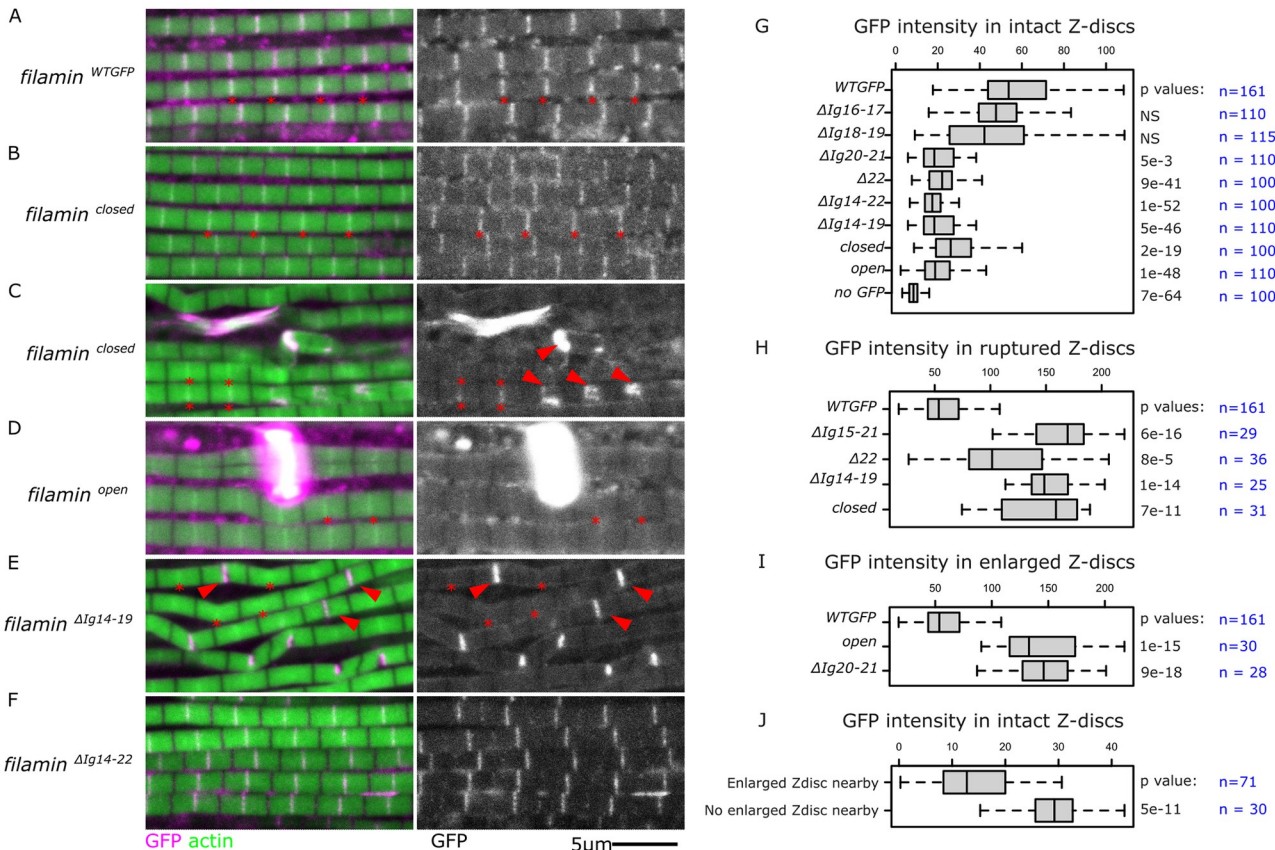

**Fig 6. The filamin mutant forms localize to the Z-disc with different intensities.** A-F) Confocal images of IFM from different homozygous filamin mutants. Actin filaments are shown in green, and GFP from the different filamin mutants in magenta. Red asterisks indicate selected Z-discs. Red arrowheads indicate ruptured discs. A) The *filamin*WT-GFP form localizes to the Z-discs (asterisks). B) The *filamin*closed-GFP mutant form localizes to the Z-disc in normal-looking myofibrils (asterisks). C) When myofibrils get damaged, the *filamin*closed-GFP mutant accumulates at the ruptured discs (arrowheads) and decreases in normal adjacent Z-discs. D) The *filamin*open-GFP mutant accumulates mostly at the enlarged Z-discs and is barely detectable in the normal Z-discs. E) The *filamin*ΔIg14-19-GFP mutant form localizes strongly in ruptured Z-discs and depletes filamin from adjacent discs. F) Removing the entire C-terminus part in *filamin*ΔIg14-22-GFP mutant form does not prevent Z-disc localization in non-ruptured myofibrils. G) Boxplot of filamin GFP intensities in different filamin homozygous mutant backgrounds. H) GFP intensities in ruptured discs in various filamin homozygous mutant backgrounds. I) GFP intensities in enlarged discs in various filamin homozygous mutant backgrounds. J) GFP intensities in intact discs in *filamin*open homozygous mutants, comparing those in proximity to enlarged discs versus those that are not. P-values were calculated using Welch's two-sample t-test followed by a Bonferroni correction. In the Boxplots, the central box represents the 25–75th percentiles, and the median is indicated. The whiskers show the minimum and maximum values.

terminal region (Fig 6H), suggesting filamin accumulation at ruptured Z-discs is mediated by the N-terminal half of filamin.

## Filamin mutants have uncoordinated wing movements

We then investigated the impact of filamin mutations on flight behavior. Damaged myofibrils do not provide as much contractile force as healthy myofibrils. Since synchronous wing movements require similar force generation from all the indirect flight muscles, and muscle damage occurs randomly, we hypothesized that filamin mutants would not be able to synchronize their wings perfectly. To measure the movement of both wings simultaneously and independently, we created a microcontroller-based device equipped with two infrared QRD1114 proximity sensors, one for each wing (Fig 7A).

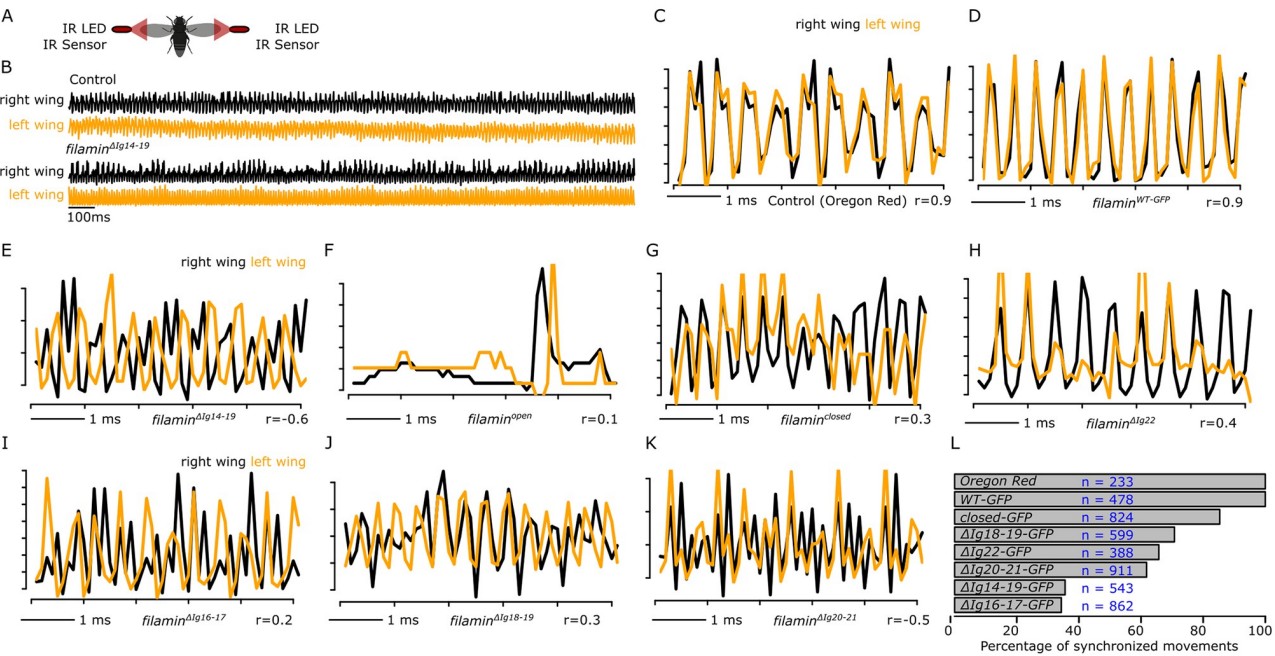

**Fig 7. Filamin mutants have less synchronous wing movements than control animals.** A) Schematic drawing of the device that simultaneously records both wing movements. Infrared (IR) LEDs coupled to very low-range light sensors detect the reflected IR light from the moving wing. B) A 1-second wing recording of control and *filamin*$^{\Delta Ig14-19-GFP}$ mutant. C) A 5-millisecond recording in a control animal. Both wings are perfectly synchronized (r = 0.9). D) A 5-millisecond recording of a *filamin*$^{WT-GFP}$ control fly. E-K) 5-millisecond recordings of different filamin homozygote mutant animals. A single wing movement is shown in F. L) Plot of the proportion of observed synchronized wing events from the entire recordings. Most events in control animals are synchronized. A decrease in the proportion of synchronized events is observed in filamin mutants. 10 flies were used for each condition and n denotes the number of analyzed events.

The sensor can detect reflected light within a range of 1 mm. As the wing moves near the sensor, it reflects the infrared light emitted by the sensor, increasing the detected signal. On the other hand, when the wings are in the fully up or fully down position, the amount of reflected light decreases. This is because the wings are no longer in the sensor's detection range. By using this technique, we were able to precisely track the movements of each wing during flight and analyze their synchronization. We first tested the two-wing sensor in Oregon Red flies. We detected very rapid oscillations of about 200 Hz in both wings (Fig 7B upper panel). A similar very fast oscillatory pattern was observed in *filamin*$^{\Delta Ig14-19-GFP}$ mutant flies (Fig 7B lower panel). Importantly, the very fast oscillations were not observed if the flies were not moving their wings. The signal obtained from the left wing and the right wing was independent of each other, as turning off the LED lights from either side only impacted that respective side. In the control Oregon Red flies, the movement of the right and left wings was perfectly synchronized as evidenced by a high Pearson's correlation coefficient (r) value (Fig 7C, r = 0.9). The *filamin*$^{WT-GFP}$ homozygote flies also had well synchronized wing movements (Fig 7D, r = 0.9). In contrast, *filamin*$^{\Delta Ig14-19-GFP}$ mutant flies have asynchronous wing movements, where the movement of one wing appears out of phase with the other wing (Fig 7E, r = -0.6).

The fast oscillations were lost in the *filamin*$^{open-GFP}$ mutants, which are flightless and cannot consistently move their wings (Fig 7F). A single double-wing movement was sometimes observed in *filamin*$^{open-GFP}$ mutants (Fig 7F). The *filamin*$^{closed-GFP}$ homozygote mutant flies had irregular wing beats with low r values (Fig 7G, r = 0.3), indicating that they struggle to

maintain wing movement synchrony. The *filamin$^{ΔIg22-GFP}$*, the *filamin$^{ΔIg16-17-GFP}$*, the *filamin-$^{Δ}$Ig18-19-GFP*, and the *filamin$^{ΔIg20-21-GFP}$* homozygote mutants all had similar irregular movements (Fig 7H–7K). To compare different genetic backgrounds, we established a threshold of r > 0.5 for synchronous movements and determined the number of synchronous events between the control and filamin mutant groups. All the flying events in both Oregon Red and *filamin$^{WT-GFP}$* control flies were synchronized (Fig 7L). In *filamin$^{closed-GFP}$* mutant flies, roughly 20% of the flying events are not synchronized. In *filamin$^{ΔIg18-19-GFP}$*, *filamin$^{ΔIg20-21-GFP}$*, and *filamin$^{ΔI22-GFP}$* mutants the number of synchronized flying events was between 60% and 70% (Fig 7L). In *filamin$^{ΔIg14-19-GFP}$* and filamin$^{ΔIg16-17-GFP}$ mutants the number of synchronized flying events was ~38%. Unsynchronized wing movements were also evident in 1-second recordings in all mutants (S5 Fig).

## Discussion

In this report we analysed the functions of filamin in maintaining indirect flight muscles using a series of C-terminal deletions and mutants affecting the opening of the MSR of filamin. All mutations that lacked the MSR or had a closed MSR led to ruptured myofibrils at Z-discs and to frayed Z-discs. In addition, filamin without a dimerisation domain led to ruptured myofibrils, but not to frayed Z-discs. Anchoring actin filaments to Z-discs (i.e. no fraying) requires the N-terminal part of filamin up to Ig14. In contrast, an open MSR and the deletion of Ig20-21 did result neither in ruptured myofibrils nor in frayed Z-discs, but in enlarged Z-discs. The N-terminal part of filamin is sufficient for the localization to Z-discs as all tested mutants localized to Z discs (i.e. MSR, rod2, dimerisation domain all not required). Interestingly, the localization to ruptured Z-discs was increased in all cases. Finally, all mutants showed defects in wing movements.

Numerous filamin variants in human patients are linked to cardiac and muscular abnormalities, yet the precise molecular mechanisms remain unclear [38]. In this study, we utilize Drosophila to conduct a comprehensive analysis of the structural and functional aspects of filamin in muscle biology. In muscles, filamin is primarily found at the Z-disc, where it interacts with actin filaments, and the presence of titin is necessary for maintaining Z-disc stability. Both mice and Drosophila lacking filamin exhibit the accumulation of ruptured myofibrils and general muscle disorganization as they age, as supported by previous studies [18,41]. Moreover, filamin is observed to accumulate at myofibril microlesions that occur during intense muscle contractions induced by either forced exercise or electrical stimulation [10,12,19]. Utilizing single-molecule localization microscopy, it has been revealed that filamin's actin-binding domains are centrally located within the Z-disc, while the dimerization domains are positioned at the periphery of the disc [23,42], providing an ideal arrangement for sensing Z-disc deformation caused by muscle contractions [22]. All these experimental data have led to the hypothesis that filamin senses mechanical damage at the Z-disc where it coordinates a compensatory stabilization mechanism that prevents the Z-disc from rupturing. Here we test this hypothesis by using well-defined filamin mutants that affect the mechanical signaling properties of filamin and by removing specific Ig domains around the mechanosensitive region of filamin. We found that filamin mechanosensing is required to prevent Z-disc rupturing during contractile load, whereas Ig 14 is required to prevent myofibril fraying.

Our comprehensive analysis revealed specific functional aspects of filamin in muscle biology: (i) filamin has a scaffolding function, which depends on Ig 14, that is crucial for anchoring actin filaments; (ii) filamin has a mechanosensitive function that is required for mechanoprotection of muscles; (iii) filamin is recruited to damaged Z-discs independent of its MSR but an open MSR is able to induce an over-recruitment of Z-discs components.

## Filamin Ig 14 is crucial for anchoring actin filaments at Z-discs

Filamin depletion through RNAi results in unstable anchoring of actin filaments at the Z-disc, leading to evident actin filament fraying in the IFM, the heart, and the larval body wall muscles [18,20,43]. Our observations show frayed myofibrils in deletions removing Ig 14 to Ig 21, but not in the deletion removing Ig 15 to Ig 21. Therefore, we have mapped the fraying phenotype specifically to the Ig 14 domain and show that fraying is independent of filamin dimerization and mechanosensation through the MSR. Mechanistically, Ig 14 may prevent actin filament fraying through the recruitment of another actin-binding protein because filamin actin binding is not required for actin filament stability. The CH actin-binding domains of filamin are unaffected in all the mutants used and an insertion mutant affecting the CH actin-binding domains does not exhibit fraying myofibrils but rather ruptured myofibrils [18].

## Filamin mechanosensing is crucial for its mechanoprotective role in muscles

Filamin serves as an elastic connection between opposing sarcomeres at the Z-disc; the C-terminal side binds to a Titin ortholog called Sallimus, and the N-terminal side to actin filaments [18,23]. Filamin maintains Z-disc cohesion either by providing elastic structural support or mechanical signaling or both. If filamin function is missing, ruptured Z-discs accumulate. We propose that filamin signals the mechanical damage of the Z-disc through conformation changes in its MSR. The MSR shuttles between closed and open conformations that engage in different signaling cascades [13,44]. At the Z-disc, the MSR can be closed resulting in no Z-disc stabilization signalling, or open resulting in signalling initiating Z-disc stabilization. We propose that contractile damage causes sarcomere overstretching which results in the mechanical opening of filamin MSR. Several pieces of evidence support our model. First, our closed filamin mutants have ruptured Z-discs, while our open filamin mutants have enlarged Z-discs which we interpret as the result of continuous Z-discs stabilization signals, suggesting that the ruptured and enlarged phenotypes represent different stages of a myofibril mechanoprotective response. Second, we found that muscle contractions are required for the rupturing of the Z-discs in the closed mutants and the mutants lacking the dimerization domain. This suggests that the closed filamin mutants do not exhibit myofibril assembly defects, but rather weakened Z-discs that eventually break under contractile load. Last, we observed ruptured Z-discs in an actin-binding filamin mutant [18]. We do not know the signaling molecules downstream of filamin opening. However, two possible proteins are Hsp70 and αB-crystallin. They both localize to muscle damage like filamin [9–11], and the *Drosophila* αB-crystallin physically interacts with filamin in muscles and causes muscle defects when removed [20]. A speculative scenario is that filamin opening recruits αB-crystallin which helps stabilize the Z-disc.

We show that the removal of the dimerization domain results in ruptured myofibrils. We propose that filamin exists as a dimer at the Z-disc, and that dimerization is required to provide an anchoring point for mechanosensation. The human FLNC dimerization domain interface is stable at pulling forces below 10 pN, slightly higher than the opening of the MSR at 4–5 pN and therefore provides stability to the mechanical function of the MSR [28]. In humans, the most common myopathy-causing FLNC variant is the W2710X mutation, the truncation allele truncating the dimerization domain. Mice carrying a FLNC W2711X mutation that removes the dimerization domain have fragile myofibrils that are sensitive to eccentric contractile damage [12,41]. Together, our data suggest a conserved role for the dimerization domain allowing mechanosensation during muscle maintenance.

## The open MSR enhances the localization of filamin to damaged Z-discs, which is mediated by its N-terminus

Filamin is one of the most used markers of myofibrillar damage in vertebrates [10,19,21]. Damaging C2C12 myotubes in culture by electrical pulse stimulation generates myofibril damage which leads to the accumulation of filamin within the first 5 minutes [10]. Both homozygous and heterozygous $FLNC^{p.W2711X}$ mutant mice, which have an early stop codon at the beginning of the dimerization domain, exhibit myofibril lesions that accumulate FLNC protein [12,41]. Here we demonstrate that filamin accumulates in ruptured Z-discs in Drosophila, suggesting that this accumulation is a conserved feature of filamin function across animals. The accumulation is prominently observed in ruptured Z-discs and is particularly pronounced in enlarged Z-discs, which deplete the filamin pools in neighboring Z-discs. We hypothesize that filamin undergoes self-recruitment to the damaged Z-discs, likely facilitated by the opening of the MSR. The purpose of this self-recruitment mechanism may be to create a positive feedback loop that ensures rapid recruitment of proteins or to replace damaged filamin molecules. With the tools currently available, we are unable to identify the specific region within filamin responsible for self-recruitment. As all our filamin mutants exhibit some level of localization to the Z-disc, we believe that a region in the N-terminal half of filamin contains a Z-disc localization sequence. Several Z-disc localization sequences may exist, which is common in Z-disc proteins.

Similar enlarged Z-disc phenotypes like the one in filamin open mutants have been shown before from a diverse set of genetic conditions and sometimes are referred to as zebra bodies. For example, by affecting the tension generated by myofibril contraction through a mutation in the motor domain in the myosin heavy chain ($MHC^{R249D}$; [45]), by reducing the amount of myosin-II, or its regulatory light chain [46], or by overexpressing truncated forms of the M-line protein Sals [47]. In addition, massively affecting transcription or translation of IFM-specific genes also leads to enlarged Z-discs [48,49]. A speculative scenario is that these conditions cause muscle hypercontraction, which indirectly results in filamin opening, therefore creating enlarged discs. Mutations in MHC indeed cause muscle hypercontraction [50].

## Filamin functions are required for synchronous muscle contractions

Flies generate wing motion through the indirect, mechanical connection between the IFM and wings, which is facilitated by the resonance of the elastic thoracic cuticle. For this resonance to work effectively, all muscles must be producing similar forces [51,52]. Filamin hypomorphic mutants $cher^{Q1415sd}$ and $cher^{A5}$ fly for the first couple of days but only for a few seconds [18].

We hypothesize that the decline in muscle function is related to myofibrillar rupturing, and to test this, we developed a device that detects wing movement of both wings simultaneously, as asynchronous movement events indicate less force generation due to the accumulation of damaged myofibrils, resulting in asymmetric muscle strengths. We observed asynchronous wing movements in all filamin mutants studied, including the 20–21 mutant with enlarged Z-discs but no ruptured Z-discs, indicating that both enlarged Z-discs and ruptured Z-discs affect muscle force generation.

## Not all filamin functions require dimerization

We show that the removal of the dimerization domain does not result in actin filament fraying. We propose that filamin's role in actin filament anchoring does not require dimerization. Other reports have also pointed out functions of filamin independent of their dimerization state. Homozygous FLNC W2711X mutant mice have relatively mild muscle defects [41].

Drosophila filamin mutants that affect the dimerization domain do not greatly affect the heart structure, whereas the depletion of filamin using an RNAi has a strong structural and functional detrimental effect on the heart [43]. Together our data suggest that some filamin functions are independent of dimerization.

## A genetic tool to dissect filamin functions in Drosophila

Apart from muscles, filamin is required for a variety of processes including the formation of ring canals in the nurse cells of the ovary [17], providing mechanical protection to nephrocytes [53], promoting macrophage infiltration during development [54], coordinating axon guidance in neurons [55], positioning the nucleus in nurse cells [56], and during epithelial tumor growth [57]. Because filamin function is linked to its binding partners, which are probably tissue-specific, we expect our mutants to help interrogate filamin functions in other tissues. Importantly, the filamin mutants we generated have different phenotypes suggesting they can be used for structure function assays.

## Supporting information

**S1 Fig. Filamin RNAi phenotype includes ruptured and frayed myofibrils.** A) Control muscles expressing Mef2-Gal4 and a Zasp52-GFP protein trap to mark the Z-discs. B and C) Examples of muscle phenotypes caused by the expression of a filamin RNAi using Mef2-Gal4. Zasp52-GFP is used to mark the Z-discs. Red arrows denote ruptured myofibrils. Purple arrows denote frayed myofibrils. The scale bar is 5 μm.
(TIFF)

**S2 Fig. Uncropped versions of the western blots presented in Fig 3.** A and B) Original western blot images corresponding to Fig 3C left panel. C) Ponceau staining corresponding to Fig 3C left panel. D-E) Original western blot images corresponding to Fig 3C right panel. F) Ponceau staining corresponding to Fig 3C right panel. The molecular weights and the genotypes are annotated in panels A and D.
(TIFF)

**S3 Fig. The frayed phenotype does not depend on damage caused during muscle contraction.** A-B) Plots of the phenotype ratios observed in the $filamin^{\Delta14\text{-}22\text{-}GFP}$ mutant muscles in flying and non-flying conditions. A) The ratio of ruptured myofibrils decreased significantly in the non-flying condition. B) The ratio of frayed myofibrils over the total number of myofibrils does not change in the non-flying condition. C) The ratio of ruptured myofibrils over the total number of myofibrils in $filamin^{\Delta22\text{-}GFP}$ mutant decreased significantly in the non-flying condition. Confidence intervals were calculated at 95 by an exact binomial test, and p-values were calculated using a 2-sample test for equality of proportions with continuity correction. D-F) Confocal image of $filamin^{\Delta14\text{-}19\text{-}GFP}$, $filamin^{\Delta14\text{-}21\text{-}GFP}$, and $filamin^{\Delta14\text{-}22}$ mutants over the $Df(3R)Ex6176$ deletion shows myofibrils that are both ruptured (red arrows) and frayed (purple arrows). G) The $filamin^{\Delta s24\text{-}attp}$ landing site mutant has a strong IFM phenotype. The top panel exhibits a mild phenotype characterized by ruptured (red arrows) and frayed myofibrils (purple arrows), while the lower panel displays a severe phenotype with similar features. In D-G, actin staining is in green, and Zasp52-mCherry is in magenta. The scale bar represents 5 μm.
(TIFF)

**S4 Fig. The enlarged Z-discs from $filamin^{\Delta20\text{-}21\text{-}GFP}$ and $filamin^{open\text{-}GFP}$ mutants are similar in size.** A) Confocal image of $filamin^{open\text{-}GFP}$ mutant muscles with Zasp52-mCherry to mark the Z-disc in green and actin stained with phalloidin in magenta. B) Confocal image of

*filamin*$^{\Delta 20-21-GFP}$ mutant muscles with Zasp52-mCherry to mark the Z-disc in green and actin stained with phalloidin in magenta. C) Boxplot of the measured area of enlarged Z-discs in both mutants. Welch two sample t-test was used for comparing the samples.
(TIFF)

**S5 Fig. Representative 1-second wing recordings of control and *filamin* mutant conditions.** Scale bar in all is 100 ms. The right-wing recording is colored black. The left-wing recording is colored orange. The overlap plot is shown, and the correlation R values are noted.
(TIFF)

**S1 Data. Complete genotypes of flies that were used in every figure and summary of phenotypes.**
(DOCX)

**S2 Data. Protein Sequence alignment of the filamin mutants used.**
(PDF)

**S3 Data. Supporting information for myofibril phenotype data, wing beat synchronization data, and GFP intensity data.**
(XLSX)

## Acknowledgments

We thank the Bloomington Drosophila stock center for materials and the Cell Imaging and Analysis Network (CIAN) imaging facility for providing access to confocal microscopy.

## Author Contributions

**Conceptualization:** Sven Huelsmann, Frieder Schöck, Nicanor González-Morales.

**Data curation:** Lucas A. B. Fisher, Nicanor González-Morales.

**Formal analysis:** Lucas A. B. Fisher, Sven Huelsmann, Nicanor González-Morales.

**Funding acquisition:** Frieder Schöck, Nicanor González-Morales.

**Investigation:** Lucas A. B. Fisher, Belén Carriquí-Madroñal, Tiara Mulder, Sven Huelsmann, Nicanor González-Morales.

**Methodology:** Lucas A. B. Fisher, Belén Carriquí-Madroñal, Tiara Mulder, Sven Huelsmann, Nicanor González-Morales.

**Project administration:** Nicanor González-Morales.

**Resources:** Belén Carriquí-Madroñal, Sven Huelsmann, Nicanor González-Morales.

**Software:** Nicanor González-Morales.

**Supervision:** Sven Huelsmann, Frieder Schöck, Nicanor González-Morales.

**Validation:** Lucas A. B. Fisher, Sven Huelsmann, Nicanor González-Morales.

**Visualization:** Nicanor González-Morales.

**Writing – original draft:** Frieder Schöck, Nicanor González-Morales.

**Writing – review & editing:** Sven Huelsmann, Frieder Schöck, Nicanor González-Morales.

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
