## [Decision Letter · Decision Letter 0]

16 Jan 2024

Dear Dr Gonzalez Morales,

Thank you very much for submitting your Research Article entitled 'Filamin protects myofibrils from contractile damage through changes in its mechanosensory region' to PLOS Genetics.

The manuscript was fully evaluated at the editorial level and by independent peer reviewers. The reviewers appreciated the attention to an important problem, but raised some substantial concerns about the current manuscript. Based on the reviews, we will not be able to accept this version of the manuscript, but we would be willing to review a much-revised version. We cannot, of course, promise publication at that time.

If you decide to revise the manuscript for further consideration at PLOS Genetics, please aim to resubmit within the next 60 days, unless it will take extra time to address the concerns of the reviewers, in which case we would appreciate an expected resubmission date by email to plosgenetics@plos.org.

We are sorry that we cannot be more positive about your manuscript at this stage. Please do not hesitate to contact us if you have any concerns or questions.

Yours sincerely,

Pablo Wappner

Academic Editor

PLOS Genetics

Gregory P. Copenhaver

Editor-in-Chief

PLOS Genetics

Reviewer's Responses to Questions

**Comments to the Authors:**

Reviewer #1: This manuscript focuses on the role of filamin in Drosophila indirect flight muscle, using genetic and cell biological approaches. The confocal images are of high quality and the quantitative approach to studying the myofibril defects is appreciated. Further, the development of a device that independently assesses each wing movement is novel and yields interesting results. Overall, it is a nice study that provides new insights into the role of filamin in myofibril structure and function. There are a number of issues that should be addressed by the authors to improve the structure and content of the manuscript.

Major Points:

1. The abstract does not provide much detail as to what was done experimentally and how the conclusions were reached. Why was the C-terminal targeted? Is it the mechanosensory region? What is novel about the mutants? Why were closed vs. open mutants tested? Does one of them prevent/enhance elastic changes? I realize that brevity is necessary in the abstract, but the various points need to be logically connected.

2. Results: Since the analysis is done in hemizygous flies, the authors should comment that only a single copy of the gene is necessary for complete rescue (is this true in regard to flight ability?). What was the reason that mutations were tested over the deletion instead of as homozygotes? On line 281, the authors state “We analyzed these mutants over a deficiency background to rule out the effect of secondary background mutations.” By this, I suppose they mean recessive mutations that might be on the mutated chromosome. Can this be explained more thoroughly? Is this the general reasoning for the study being done over the deletion? Why mention it only on line 282?

3. Results: Is it not unexpected that the filamin mutants with ruptured Z disks are able to fly? Quantitative flight tests seem an important missing element in the paper, but this is largely made up for by the novel wing beat analysis. Perhaps this should be mentioned earlier, i.e., the flight abilities are quantitatively assessed at the end of the Results.

4. Line 186: the eccentric contractions referred to here are in human muscles. Are you implying that you are testing for eccentric contractions as being causative in Drosophila? Is there evidence for eccentric contractions and resulting myofibril defects in Drosophila?

5. Figure 3C: The left panel may be problematic, as it appears to contain multiple blots spliced together.

6. The conclusion regarding the origin of enlarged Z-discs is speculative. See the following paper wherein multiple references are given for IFMs that display enlarged Z-discs under various genotypes/conditions: https://doi.org/10.3390/biology11081137 . These other instances should at least be mentioned and/or argued to fit (or not fit) with the authors’ speculations.

7. The localization of filamin in the various filamin mutants is interesting, but without quantification, can the authors really conclude that levels are reduced in the normal sarcomeres of mutants that have enlarged Z-discs? At the least, it would be helpful to state (assuming this is the case) that the confocal images in the mutants are taken at the same settings as the control.

8. Line 394: “All these experimental data have led to the hypothesis that filamin senses mechanical damage at the Z-disc where it coordinates myofibril repair. Here we test this hypothesis…” I see no evidence in the manuscript that filamin coordinates myofibril repair or testing of this issue. In fact, it is unclear that such repair occurs in Drosophila IFM. The requirement of filamin to prevent Z-disc rupturing or myofibril fraying likely indicates it plays a role in maintaining stability, rather than in repairing the defects, i.e., “mechanoprotection”, as mentioned by the authors.

Minor Points:

1. Abstract: “either by providing elastic support or through signaling.” Perhaps this should be: “by providing elastic support and/or through signaling.”

2. Introduction: “Striated muscles are made of myofibrils and repeated sarcomeres.” There are other constituents as well, e.g., mitochondria, nuclei.

3. Introduction: Confusing sentence should be reordered: “During this process, some Z-disc proteins accumulate at the lesions and are thought to mediate Z-disc repair, including Filamin, Xin, Hsp70, and αB-crystallin [7][8][9].” Change to: “During this process, some Z-disc proteins, including Filamin, Xin, Hsp70, and αB-crystallin [7][8][9], accumulate at the lesions and are thought to mediate Z-disc repair.”

4. Results: “Because of the suspected role of filamin in muscle maintenance, we used 15 day old flies”. Authors need to explain the connection between maintenance and the age of the flies, e.g., 15 days is well after myofibrils assembly is complete.

5. Figure 1D legend: ‘big’ and ‘enlarged’ need not both be used to describe the phenotype.

6. Line 241 statement is unclear: We noted that the appearance of ruptured myofibrils decreased in flies prevented from flying but the ratio of frayed myofibrils remained constant between flying and non-flying animals

7. Figure 4: Mention that 14-21 mutant myofibrils are not shown. Do the flies mentioned in this figure fly?

8. Line 356: Is there a rationale for choosing r > 0.65 as the threshold for synchronous wing movement?

9. Line 484: “we expect our domains to help interrogate filamin functions in other tissues”. Unclear what “our domains” refers to here.

10. Figure S1 Legend. The title is confusing perhaps instead of “contractile damage”, use “damage caused during muscle contraction”.

Reviewer #2: General comments:

This paper presents a well done and interesting analysis of how mutations in the Ig-like domains of filamin affect myofibril structure in the Drosophila IFMs. The authors make use of previously published filamin-GFP mutants that affect the ability of the mechanosensor region (MSR) to operate, presumably in response to mechanical forces. Two different phenotypes are observed in open or closed filamin mutants: ruptured myofibrils or enlarged Z-discs. Further genomic engineering approaches to generate different C-terminal domain deletions narrowed down the Ig 14 domain of filamin as essential for anchoring actin filament at Z discs. The authors also show that muscle contraction is required to induce Z-disc rupture, thereby confirming previous reports that the MSR domain of filamin is required for force transmission. In this sense, the current manuscript largely confirms previous findings rather than uncovering new mechanisms that contribute to filamin function.

ESSENTIAL COMMENTS

• The authors state that Z-discs are enlarged in open filamin mutants. This is indeed true for the Z-disc protein Zasp52 and actin. Are other Z-disc proteins also enriched to substantiate this statement? Or is this recruitment specific to Zasp52? Also, the localization of Zasp52 and F-actin are different (see panel 1D and others) as actin seems to be broader. What is the significance of this?

• Lines 82-83 in the last paragraph of the introduction reads ‘We found that the Ig domain pairs 16-17 and 18-19 are required for Z-disc repair.’ This seems to be an overstatement as there are no experiments looking at repair. This statement should be supported with experimental evidence or removed.

• The authors have previously shown that 4 isoforms of filament are present in the Drosophila thoraces (PMID: 28732005). There is no discussion of which isoforms are tagged in the current GFP constructs. How do these additional isoforms affect the interpretation of filamin mutants in the present manuscript since the protein is known to dimerize? Moreover, how do the authors reconcile the specific Z-disc staining of Cher-FLAG (PMID: 28732005) vs the partial Z-disc plus cytoplasmic localization of filamin in Figure 6? Is this an isoform issue?

• How many copies of filamin-GFP mutants are in used in Figures 1-5? Figure 6 specifically mentions homozygous filamin, but it is not clear in the other figures. Obviously filamin-GFP mutants/Df give rise to adults, but is there partial lethality associated with any of the mutants. It seems like this is a possibility considering the importance of filamin during development and the severe consequences upon manipulation of the Ig domains.

• Quantitation of the phenotypes in Figure 1 should be included. It is not clear how prevalent these defects are (in 5% of myofibrils? 50%). Maybe a low mag picture would help clarify this issue. More information also needs to be included concerning quantification and statistics. The authors could include this in the methods section. Also include sample size for all graphs. Maybe this is included in Data S2, but this is not mentioned anywhere in the text.

• The conclusion in panel 2D (enlarged Z-discs are not affected in filamin open mutants) is not convincing as shown. While it is true that the flying vs non-flying appear may not reach statistical significance levels, there is a substantial difference/trend between these two data sets that needs to be clarified. Maybe more sample size or a different type of analysis is required? Also please clarify what exactly the y-axis (ratio of affaected myofibrils) indicates. Ratio of affected to unaffected? Why not plot this as % affected?

• The Western blots in Figure 4C should include a loading control as the levels of filamin are not equal in different mutants. Are some filamin mutants less stable or degraded at higher levels? Do the lower levels of protein correspond to phenotypes?

• Line 237, “Finally, we tested filamins24-attp, the original landing mutant, which removes the entire region starting from Ig 14 and lacks a stop codon. Rupturing Z-discs and fraying myofibrils are common in the filamins24-attp mutant muscles.” There is no picture in Figure 4 to support this sentence.

• There is no control muscle picture in Figure 4. It is not clear why some mutants are repeated multiple times in Figure 4. Is this to show different phenotypes? The graphs in panels F-H contain no statistics or sample size information. Also lines 245-6 seems like an overstatement without additional data. Please modify.

• Lines 275-7. The mutations in the main text do not match the figure panels, so please check. IG 16-17 and IG 18-19 are in C and D of Figure 5. Also please include quantitation of filamin-closed as it is mentioned to be the same (line 275).

• Supplemental Figure S1 should be included as primary data.

• The introduction discusses the recruitment of z-disc proteins and the repair of Z-discs associated with Xin, Hsp70 and CryAB. Does filamin interact with these proteins or has an association to cause muscle defects? If any of these ideas have merit there could be some discussion of them. There is an extensive discussion of specific possibilities, but a few general possibilities could be presented too.

OTHER COMMENTS

• Lines 173-4 state that filamin open mutants are flightless but closed are not. This data should be shown here or this statement should be moved elsewhere.

• Line 217 is the first mention of the S24 mutation in the body of the text, while the explanation of the mutation is mentioned in the Fig3 legend. Maybe the introduction of S24 is better understood and easier to read in the main text.

• Line 220 – remov

---

## [Decision Letter · Decision Letter 1]

10 Apr 2024

Dear Dr Gonzalez Morales,

Thank you very much for submitting a revised version of your Article entitled 'Filamin protects myofibrils from contractile damage through changes in its mechanosensory region' to PLOS Genetics.

The manuscript was evaluated the same three reviewers that read the original version. They found that the manuscript has very much improved but still needs to be modified in order to be accepted for publication. Please pay special attention to the comments of reviewer 3, but of course to the other two reviewers as well.

We therefore ask you to modify the manuscript according to the reviewers recommendations. Your revisions should address the specific points made by each reviewer.

Yours sincerely,

Pablo Wappner

Academic Editor

PLOS Genetics

Gregory P. Copenhaver

Section Editor

PLOS Genetics

Reviewer's Responses to Questions

**Comments to the Authors:**

Reviewer #1: The authors have done an excellent job in responding to my comments and I have only one minor concern. The use of the term "heterozygote" to refer to a mutation over a deficiency may be confusing to the reader. I suggest including something like: "heterozygote (mutation/deficiency)" or "hemizygote" to better clarify this.

Reviewer #2: In many aspects the manuscript is improved. Below are some items that need another look or were not addressed by the authors in their response.

(1) The genotypes in S3 Data do not match the panels in Figure 1, Figure 3, or Figure 7.

(2) The response to reviewer states that ‘We have incorporated the percentage of affected myofibrils alongside the images in Figure 1.’ That data is not present. The only graphs in Figure 1 are flight data.

(3) Sample size is not included for graphs in Figure 1 or Figure 6. This data is also not present in the Figure legends as requested.

(4) Maybe the red text on graphs can be changes to a different color. It is difficult to read as shown.

(5) The authors did not provide a response to the following query in the original review-

The conclusion in panel 2D (enlarged Z-discs are not affected in filamin open mutants) is not convincing as shown. While it is true that the flying vs non-flying appear may not reach statistical significance levels, there is a substantial difference/trend between these two data sets that needs to be clarified. Maybe more sample size or a different type of analysis is required?

(6) The authors state ‘We have included loading controls in the form of Ponceau staining on the blots in Figure 4C, which is now presented as Figure S4.’ This not correct as the blots are Fig. S2.

(7) The following statement was not addressed by the authors - It is not clear why some mutants are repeated multiple times in Figure 4. Is this to show different phenotypes?

(8) The authors did not provide responses to the following queries in the original review-

• Lines 275-7. The mutations in the main text do not match the figure panels, so please check. IG 16-17 and IG 18-19 are in C and D of Figure 5. Also please include quantitation of filamin-closed as it is mentioned to be the same (line 275).

• Supplemental Figure S1 should be included as primary data.

• The introduction discusses the recruitment of z-disc proteins and the repair of Z-discs associated with Xin, Hsp70 and CryAB. Does filamin interact with these proteins or has an association to cause muscle defects? If any of these ideas have merit there could be some discussion of them. There is an extensive discussion of specific possibilities, but a few general possibilities could be presented too.

Reviewer #3: While the revised manuscript is improved, several issues remain and are listed below. In addition, the writing was could be significantly improved.

Abstract:

1. Lines 19-21: Authors write “We made novel filamin mutations affecting the C-terminal region to interrogate the mechanosensitive region and detected two Z-disc phenotypes: dissociation of actin filaments and Z-disc rupture”. However, isn’t the enlarged Z-disc observed in the deletion of Ig20-21 also a phenotype? We acknowledge it is not the MSR region, but that deletion does, indeed, affect the C-terminal region.

2. On line 26, we suggest inserting “stabilizes” before “and counteracts contractile damage at the Z-disc”, as it is a key aspect discussed in the paper.

Introduction:

3. Line 48: Authors write “The Z-disc is a thin structure that bisects the sarcomere”. This statement is inaccurate, the Z-disc defines the boundaries of the sarcomere, it does not bisect/divide the sarcomere in two parts.

4. Line 71: Please correct typo “Drosophila” instead of “Drosophia”

5. Line 92: Authors state “enlarged Z-discs, which we interpret as a compensatory mechanism to stabilize the Z-disc and prevent it from breaking”. It is unclear, however, what this a compensatory mechanism is for. Please specify whether this is a compensatory mechanism for eccentric contractions and/or sarcomeric damage and/or both, if that is indeed what authors mean.

Methods:

6. Please specify that Ponceau staining was used as a loading control and the specifics regarding that stain.

7. In the response to reviewer 3, point 8c, authors indicated that in order to calculate the ratios of affected myofibrils, they used uncropped images each containing approximately 25 myofibrils +/-2. However, in lines 145-146, authors state that to calculate ratios of affected myofibrils, they used uncropped images typically containing 10-15 myofibrils. Please clarify this discrepancy.

Results:

8. Line 197: Add “Fig. S1” next to reference 16.

9. Overall comment for figures 4, 5, 6 and 7: Please keep a consistent order for the genotypes both in the IF images and corresponding plots. The current inconsistency in the order makes it very confusing to follow, as the order changes within the same figure and across figures.

Figure 1:

10. Indicate in panels A and B that the contractile forces are referring to eccentric contractile forces.

11. Panel B needs more clarity and specificity. We suggest adding more details (i.e.: indicate where the Z-disc is, and clear labeling N-terminal and C-terminal regions of filamin).

12. Modify colors of panels A and B so that they match the staining from the IFs (i.e.: highlight Z-discs in panel A in magenta instead of green; in panel B, use a different color for filamin, as green can be confusing given the IF stainings from the panels below, which also use green for actin).

13. Please describe in the methods how the “Percentage that flew” was assessed. Also, we suggest changing the axis as “Percentage of flying adult Drosophila” for better clarity.

Figure 6:

14. Please describe in the methods the specifics on how the GFP intensity quantification from Figure 6 was performed (e.g.: line plots across the Z-disc? Mean gray value within an area? How was the area selected? etc).

Figure 7:

15. Panel D still shows the wrong genotype, please change for WT-GFP.

16. Lines 320-322: The statement “The filamin closed-GFP homozygote mutant flies had irregular wing beats with low r values (Fig 7G, r = 0.3), indicating that they struggle to maintain wing movement synchrony” conflicts with Figure 7L, which shows 80% synchrony. This raises an issue in how the parameters (i.e.: synchronicity and correlation) are defined in the text and needs clarification. We understand the Pearson’s correlation coefficient is the indicator of a match between wing left and wing oscillation amplitude, however, it is important to note that even if the oscillations might not have the same amplitude (r=0.3), they might still exhibit high synchronicity (80% of events). Line 315 should be revised to provide better understanding in how to interpret the Pearson’s correlation coefficient, and the text in the results sections should be adjusted accordingly to reflect these interpretations.

Figure S3:

17. Please add p-values/stats in panel C, as well as “n” in panels A, B, and C.

S3 Data:

18. Please change the letters to match the panels in Figure 1 (i.e.: the first genotype should correspond to the letter D, and so on, until the letter J).

19. Genotypes denoted in Figure 7 of the Word document do not correspond with the order of notation in the blots, please rearrange Word genotypes in order to keep the same order as in the blots.

20. Edit panel names for Figure 7 in word document to match the figure panel letters.

**Have all data underlying the figures and results presented in the manuscript been provided?**

Reviewer #1: Yes

Reviewer #2: Yes

Reviewer #3: None

PLOS authors have the option to publish the peer review history of their article (what does this mean?). If published, this will include your full peer review and any attached files.

Reviewer #1: **Yes: **Sanford I. Bernstein

Reviewer #2: No

Reviewer #3: No

---

## [Decision Letter · Decision Letter 2]

7 Jun 2024

Dear Dr González Morales,

We are pleased to inform you that your manuscript entitled "Filamin protects myofibrils from contractile damage through changes in its mechanosensory region" has been editorially accepted for publication in PLOS Genetics. Congratulations!

Yours sincerely,

Gregory P. Copenhaver, Ph.D.

Section Editor

PLOS Genetics

Comments from the reviewers (if applicable):

Reviewer's Responses to Questions

**Comments to the Authors:**

Reviewer #2: Comments have been addressed.

**Have all data underlying the figures and results presented in the manuscript been provided?**

Reviewer #2: Yes

PLOS authors have the option to publish the peer review history of their article (what does this mean?). If published, this will include your full peer review and any attached files.

Reviewer #2: No

**Data Deposition**

http://datadryad.org/submit?journalID=pgenetics&manu=PGENETICS-D-23-01361R2

**Press Queries**

---

## [Editor Report · Acceptance letter]

19 Jun 2024

PGENETICS-D-23-01361R2 

Filamin protects myofibrils from contractile damage through changes in its mechanosensory region 

Dear Dr González Morales, 

We are pleased to inform you that your manuscript entitled "Filamin protects myofibrils from contractile damage through changes in its mechanosensory region" has been formally accepted for publication in PLOS Genetics! Your manuscript is now with our production department and you will be notified of the publication date in due course.

With kind regards,

Judit Kozma

PLOS Genetics

On behalf of:
